# Reproductive Behavior and Development of the Global Insect Pest, Cotton Seed Bug *Oxycarenus hyalinipennis*

**DOI:** 10.3390/insects15010065

**Published:** 2024-01-17

**Authors:** Ahmed M. Saveer, Jing Hu, Jaime Strickland, Robert Krueger, Shannon Clafford, Aijun Zhang

**Affiliations:** 1Invasive Insect Biocontrol and Behavior Laboratory, Beltsville Agricultural Research Center-West, USDA-ARS, Beltsville, MD 20705, USA; jing.hu@usda.gov (J.H.); jaime.strickland@usda.gov (J.S.); 2National Clonal Germplasm Repository for Citrus & Date, USDA-ARS, Riverside, CA 92507, USA; robert.krueger@usda.gov; 3Orange County Agricultural Commissioner’s Office, Orange, CA 92865, USA; shannon.clafford@ocpw.ocgov.com

**Keywords:** *Oxycarenus hyalinipennis*, invasive species, longevity of adults, egg development, metamorphosis, mating behavior

## Abstract

**Simple Summary:**

The cotton seed bug, *Oxycarenus hyalinipennis*, is an invasive pest native to southern Europe and North Africa. It feeds on plants in the mallow family, including the agricultural commodities cotton, okra, hibiscus, cocoa, and kenaf. The *O. hyalinipennis* is a major economic threat to the U.S. cotton industry. Based on host and climate availabilities and the lack of natural enemies, *O. hyalinipennis* has a high likelihood for establishment throughout the southern regions in the U.S. The fundamental life history and reproductive behavior of *O. hyalinipennis* have been inadequately explored until now; therefore, it is difficult to develop corresponding precautionary measures. In this research, we have conducted a comprehensive study on the life cycle of *O. hyalinipennis*, including egg development, metamorphosis, adult mating behavior, adult lifespan, and survivorship. Our results provide valuable information for scientists and growers to develop efficient management strategies in IPM for timely infestation intervention to protect the cotton crop and other agricultural commodities from *O. hyalinipennis* damage.

**Abstract:**

Understanding the fundamental life cycle and reproductive behavior of a pest insect is essential for developing efficient control strategies; however, much of this knowledge remains elusive for a multitude of insects, including the cotton seed bug, *Oxycarenus hyalinipennis*. Here, we report the results of our comprehensive study on the cotton seed bug’s life cycle, including mating behavior, adult lifespan, and egg-to-adulthood development. Our findings showed that adult males and females began mating as early as three days after emerging (75%), and the frequency of mating increased to 100% by the fifth day. Mated females commenced oviposition on cotton seeds as early as two days after mating, with a cumulative mean number of 151 fertile eggs oviposited during the first oviposition cycle. Furthermore, around 10% of eggs from both mated and unmated females remained unfertilized. The first instar nymphs began emerging approximately seven days following oviposition. To track their development, we monitored the newly hatched nymphs daily until they reached adulthood. There were five nymphal stages, which cumulatively took roughly 28 to 30 days. Notably, mating positively influenced the survivorship and lifespan of adult *O. hyalinipennis*. Mated males and females exhibited median lifespans of 28 and 25 days, respectively. In contrast, unmated males and females only lived for a median lifespan of 9.5 days, about one-third that of the mated *O. hyalinipennis*. Our study provides key insights into the *O. hyalinipennis* life history for new IPM strategies.

## 1. Introduction

Insects exhibit a wide range of reproductive behaviors that empower them to flourish in various environments. These behavioral traits possess innate adaptability, responding to both internal and external cues, thereby bolstering individual fitness. However, these behavioral characteristics can also be harnessed for pest monitoring and controlling purposes [1]. Therefore, comprehending the reproductive behavior and basic life history of pests has become indispensable in developing successful integrated pest management (IPM) strategies. These strategies aim to proactively prevent the emergence of pest issues, while also optimizing interventions to achieve maximal effectiveness and minimize expenses [2,3].

One such pest species is the cotton seed bug, *Oxycarenus hyalinipennis* (Costa) (Heteroptera; Lygaeidae). It is a significant pest that affects seeds of cotton, *Gossypium herbaceum *Linnaeus, and other malvaceous crops globally. cotton seed bugs undergo incomplete metamorphosis, also known as paurometabolous development, with three distinct life stages: egg, nymph, and adult. As cotton bolls open, the cotton seed bug adults become active and start feeding, mating, and laying eggs. Nymphal development occurs within the seed bolls. Full generation is completed in approximately one month. Depending on environmental factors, such as host availability and temperature, there can be four to seven generations in a year. Feeding adult and nymph CBSs can result in weight loss in cottonseeds, hinder seed germination, and diminish oil yield. Notably, cotton seed bugs can also be found on plants from the Tiliaceae and Sterculiaceae families [4]. Originally from Africa, cotton seed bugs have rapidly expanded their distribution to encompass regions in Asia, Europe, the Middle East, South America, and Caribbean [5]. During outbreaks in Israel, this insect has been observed aggregating on a variety of trees and shrubs, such as palms, fig trees, avocado trees, and persimmon trees, causing severe damage to non-malvaceous fruits and seeds [6]. Adults and nymphs commonly aggregate in tight clusters, especially in seed pods or bolls that emit a pungent odor. They suck oil from mature seeds and fluids from the leaves of young stems to obtain moisture [7]. Various control techniques within IPM strategies have been utilized to combat cotton seed bugs, encompassing chemical control [8], insect growth regulators [9], cultural control [10], and biological interventions [11]. Furthermore, cotton seed bugs have developed resistance to numerous commonly used insecticides [8]. More recently, cotton seed bugs invaded the Florida Keys, but were successfully eradicated in 2014. In 2019, the bug was detected near cotton fields in California, raising concerns about its potential establishment in the area. The southern regions of the United States are particularly susceptible to cotton seed bug establishment, due to favorable host availability and climate conditions, coupled with the absence of natural predators. Therefore, the risk of cotton seed bugs being introduced to the United States (U.S.) is significant. Despite the economic risks posed by cotton seed bugs, their biology, and reproductive behaviors have not been subjected to detailed study.

Here, we conducted detailed studies of the general life history parameters and reproductive behaviors of the cotton seed bug. Our goals were (1) to determine sexual maturity metrics including copulation latency, frequency, and duration, (2) to establish the egg-laying pattern of both unmated and mated females, (3) to gauge the developmental timeline of cotton seed bug from egg to adult reared under standardized laboratory conditions, and (4) to establish the impact of mating on male and female longevity.

## 2. Materials and Methods

### 2.1. Insects

Cotton seed bugs were originally obtained from primrose street trees, *Lagunaria patersonia*, infested with *O. hyalinipennis* in Irvine, Orange County, California, on 24 October 2021 (Coordinates of the location: 33° 2′ 8″ N 117° 46′ 23″ W). Primrose trees yield small brown fruits that resemble cotton bolls, which are shed during autumn. The seeds inside the shells serve as the primary food source for cotton seed bugs.

After being collected in California, the adult cotton seed bugs were held in the quarantine room at our laboratory, the U.S. Department of Agriculture (USDA), Agricultural Research Service (ARS), Invasive Insect Behavior and Biological Control Laboratory (IIBBL). The adult cotton seed bugs were reared in a transparent plastic rectangular container (42 oz, Pioneer Plastic, North Dixon, KY, USA) with a decorative inset lid. The container was constructed by drilling four 0.5-inch diameter holes in each of the walls and a single 0.5-inch diameter opening at the center of lid. These openings were fitted with a nylon screen (0.3 mm mesh size) affixed securely with hot glue, promoting proper air circulation. cotton seed bugs in the container had access to dry 15–20 cotton seeds (Cotton Fiber Bioscience Research Unit, USDA-ARS, New Orleans, LA, USA), approximately 100 g of green beans *Phaseolus vulgaris* (MOM’s organic market, College Park, MD, USA), and tap water, which was provided in a 15 mL glass bottle equipped with a cotton wick; the water and green beans were replaced as needed (3–5 days). Every 3 to 4 days, the cotton seeds containing eggs in the adults’ container were moved to a new container with green beans and water. Simultaneously, the adult container was replenished with fresh cotton seeds. In this way, adults and nymphs were reared in separate containers, each labeled with their corresponding dates, and maintained in a controlled environment chamber at a temperature of 26 °C ± 1 °C and 75% relative humidity (RH), and a 16 h light/8 h dark cycle.

### 2.2. Experimental Design and Procedure

#### 2.2.1. *Oxycarenus hyalinipennis* Copulation: Frequency, Latency, and Duration Analysis

To obtain unmated males and females for courtship behavior experiments, fifth instar nymphs were individually placed into glass vials (7.5 mL). Adult unmated males and females were collected on the day of eclosion and grouped in a separate polystyrene wide-mouth threaded container with a screened side and top, and provided beans, water, and 2–3 cotton seeds until the commencement of experiments. To measure the copulation behaviors, a single unmated female and an unmated male of the same age were paired in a mating arena (90 mm Petri dish). Three mating related behaviors were recorded: (1) copulation latency, i.e., the time taken to initiate the mating; (2) copulation frequency, i.e., the total number of pairs that initiated mating within one hour; and (3) copulation duration, i.e., the time required to finish each copulatory event. We visually recorded mating behaviors using a stopwatch for timekeeping. Each mating pair was observed for a minimum of one hour to note copulation latency and frequency. Additionally, the mating recording extended until physical separation to measure mating duration. Mate was scored when a male maintained copulation for at least 1 min. On a few pairs, copulation duration was not recorded as they maintained copulation overnight. Mating experiments were conducted 0, 1, 2, 3, 4, and 5 days post emergence (*n* = 20 per age group). Each *O. hyalinipennis* was used only once in the bioassay, and all the experiments were performed between 4 and 9 h into the photophase.

#### 2.2.2. Egg-Laying Patterns in Virgin and Mated Female *O. hyalinipennis*

To determine whether unmated females laid eggs, we conducted daily observations on newly emerged unmated females from their emergence until they ceased laying eggs for 10 consecutive days. We placed individual newly emerged unmated females in 2 oz clear straight-sided polystyrene jars with white polypropylene screw caps (Thermo Fisher Scientific, Waltham, MA, USA). The jar was constructed by drilling two 0.5-inch holes in the wall, located opposite to each other, and one 0.5-inch hole in the center of the lid. All these holes were fitted with 0.3 mm nylon screens securely attached using hot glue to ensure proper air circulation. Each jar contained a single cotton seed used as an egg-laying substrate, a 4–5 cm piece of green bean, and a moist dental cotton wick. Every day, we placed a fresh cotton seed into the plastic container and moved the cotton seed with eggs to a new 7.5 mL glass vial to track egg counts and monitoring hatching using a microscope for up to 10 days post egg-laying.

To determine the number of fertile and infertile eggs laid by a mated female, we paired a 5-day-old unmated female with an unmated male of the same age and kept them together for 24 h to ensure mating, as some females engaged in prolonged mating. After 24 h, we removed the male, and the female was provided with a cotton seed, a piece of green bean (4–5 cm long), and a moist dental cotton wick. We transferred the cotton seed with eggs to a new clean 7.5 mL glass vial to track the egg count and hatching rate, while providing a fresh cotton seed for oviposition, as previously described. This procedure was repeated daily until the female ceased laying eggs for ten consecutive days. We examined the cotton seed with eggs every day for ten days post egg-laying to detect the presence of eggs and recorded the number of fertile and infertile eggs under a microscope. Each female was used only once in the bioassay (*n* = 10–13). The green beans and water source were replaced every other day to prevent mold growth.

#### 2.2.3. Development of *O. hyalinipennis* Life Stages after Egg-Laying

In order to determine the life cycles of *O. hyalinipennis*, eggs were collected by placing cotton seeds in the container housing the adults, which consisted of approximately 60 females, for 24 h. The eggs were then collected by detaching from the cotton seeds under a microscope. Each *O. hyalinipennis* egg was placed individually in a FALCON Petri dish (35 × 10 mm, Corning Incorporated, Corning, NY, USA) along with a wet dental cotton wick. Upon hatching, the first instar nymph was transferred to a larger FALCON Petri dish (60 × 15 mm, Corning Incorporated, Corning, NY, USA) containing a cotton seed, a piece of green bean, and a wet dental cotton wick. The containers were frequently checked and replaced when mold appeared. The duration of each developmental stage from egg to adulthood (in days) was recorded, and photographs were taken as records.

#### 2.2.4. Longevity of Adult Male and Female *O. hyalinipennis*

To determine the lifespan of adult males and females, we collected unmated individuals on the day they emerged and subjected them to one of two experimental conditions. (1) Unmated males and females of the same age were placed in separate Petri dishes (35 × 10 mm, Corning Incorporated, Corning, NY, USA), each containing a water-soaked wick, a small piece of green bean, and a cotton seed. They remained in these dishes until their demise (unmated males: *n* = 30, unmated females: *n* = 30). (2) Each unmated male and female of the same age were housed together in a Petri dish to allow mating. After mating, they were separated and placed in individual dishes, each with a water-soaked wick, a small piece of green bean, and a cotton seed. They were observed until they expired (mated males: *n* = 30, mated females: *n* = 30). The green beans and water were replenished as needed throughout the duration of the experiment.

### 2.3. Data Analysis

Non-parametric Kruskal–Wallis tests followed by Dunn’s post hoc tests were used to compare copulation latency, frequency, and duration between different age groups. In all statistical analyses, the significance level (α) was set to 0.05. The Kaplan–Meier method was employed to create survival curves from the raw data and a Gehan–Breslow–Wilcoxon test was used to compare the survival curves of unmated and mated males and females. Statistical analysis was performed in GraphPad Prism v.9.5.0 (trial version).

## 3. Results

### 3.1. Age-Dependent Effects on O. hyalinipennis Copulation Behavior

To evaluate copulation metrics, including latency, frequency, and duration, we placed age-matched unmated males and females individually in a Petri dish, observing them for an hour. The age of both males and females had a notable impact on the time it took to initiate copulation (copulation latency). Mating behavior was absent in newly emerged, one-day-old, and two-day-old males and females (Figure 1a). Among the adult population, three-day-old individuals exhibited a significantly extended period of latency before beginning copulation, with a mean duration of 14.20 min (Kruskal–Wallis (Dunn’s multiple comparison): *p* < 0.0001; Figure 1a). While there was not a significant difference between four-day-old and five-day-old adults, the five-day-old individuals exhibited the shortest mean latency of 3.30 min, in contrast to the 4-day-olds, who exhibited a mean latency of 5.26 min before initiating mating (Figure 1a).

Mating initiated starting from day 3, with over 75% of unmated males and unmated females successfully engaging in mating after displaying courtship behaviors such as approach, contact, mounting, and eventually mating. As age progressed, both males and females demonstrated a strong positive effect on the mating frequency. Notably, 95% of four-day-old and 100% of five-day-old males and females achieved successful mating events (Figure 1b). Finally, the duration of copulation remained unaffected by age, as evidenced by mean durations of 98.31, 126, and 104 min for individuals aged 3, 4, and 5 days, respectively (Figure 1c).

### 3.2. Egg-Laying Patterns in Unmated and Mated Females

To better understand the egg-laying behavior of unmated and mated female *O. hyalinipennis*, we monitored individual females of both statuses throughout their first egg-laying cycle since emergence and first mating until females ceased egg-laying. Unmated females initiated egg-laying on day 7 and ceased on day 29 post emergence. Although we monitored unmated females until day 40, none of the females laid eggs after day 29. The cumulative mean number of eggs oviposited per female was 14.2 (Figure 2a). None of the eggs oviposited by unmated females were fertile. On the other hand, mated females started laying fertile eggs from day 2 and stopped on day 31 post mating. The cumulative mean number of fertile eggs oviposited during the first oviposition cycle (after mating for 24 h) was 151 eggs per female (Figure 2c). Interestingly, approximately 10% of eggs oviposited by the mated females were infertile (cumulative mean was 15.53 eggs per female) (Figure 2b) and mated females started laying infertile eggs from day 14 and stopped on day 34 post mating (Figure 2b).

### 3.3. Development of O. hyalinipennis Life Strategy

We monitored the developmental stages of cotton seed bugs, spanning from egg to adulthood, to determine developmental timelines under standardized laboratory conditions. Following mating, females commenced the process of oviposition, with an average duration of 5.3 days. Subsequent to egg deposition, the emergence of first instars occurred with an average developmental time of 6.92 days (Figure 3). The developmental durations for the second, third, fourth, and fifth instars were not significantly different, with averages of 4.12, 4.04, 3.86, and 3.92 days, respectively. The mean developmental period from the fifth instar to the emergence of adult cotton seed bugs was 6.1 days (Figure 3). Overall, the cumulative developmental timeline spanning from egg to adult encompassed 28.96 days (Figure 3). We also monitored unmated females to determine whether they laid eggs and tracked their developmental progress. Although 40–50% (*n* = 10) of unmated females did initiate egg-laying 3 to 4 days post emergence, none of the eggs hatched.

### 3.4. Longevity of Adult Male and Female Cotton Seed Bugs

To gain deeper insights into how mating affects the lifespan of adult male and female cotton seed bugs, we conducted individual observations starting from the moment of their emergence until the end of their lives. Mating had a significant impact on the survival of adult cotton seed bugs. Females that engaged in mating exhibited a noteworthy median survival period of 25 days, in stark contrast to the mere 9.5 days observed for unmated females (Gehan–Breslow–Wilcoxon test: *df* = 1; hazard ratio (log-rank): 0.7472; *p* < 0.0003; Figure 4a). A similar trend was evident among males: mated males displayed a median survival span of 28 days, while their unmated counterparts survived for only 9.5 days (Gehan–Breslow–Wilcoxon test: *df* = 1; hazard ratio (log-rank): 0.5362; *p* < 0.0038; Figure 4b).

## 4. Discussion

The cotton seed bug *O. hyalinipennis* feeds on seeds to complete its life cycle and is an invasive pest on cotton and other economically important crops. Adult male and female cotton seed bugs are expected to monitor their evolving internal reproductive physiology and exhibit specific behaviors in an adaptive coordinated manner that supports their maximum reproductive fitness. Here, we demonstrate that the reproductive behavior of *O. hyalinipennis* is strongly associated with the chronological age of adult males and females after their emergence. Neither males nor females displayed any pre-copulatory courtship behaviors until the second day after emerging as adults, during which period both adults remained sexually inactive (Figure 1a–c). This could be attributed to the rate at which individuals can generate gametes. A similar delay in the onset of reproductive behaviors has been observed in the case of the Western tarnished plant bug, *Lygus hesperus* [12]. Pre-copulatory courtship behaviors, such as approach, physical contact, mounting, and attempted mating, commenced on the third day after adult emergence. This timing coincided with the onset of copulation, during which 75% of the three-day-old adults successfully engaged in copulation (Figure 1b). Notably, individuals aged three days old exhibited the longest copulation latency (Figure 1a). This extended latency was attributed to both males and females spending more time evaluating each other and undergoing several mating attempts before the actual initiation of copulation. However, copulation latency experienced a decline starting from the fourth day of emergence, reaching its shortest duration on the fifth day of emergence. Similar age-related courtship latency has been observed in male *D. melanogaster*, wherein younger males take significantly longer than older males to initiate courtship [13]. Conversely, the frequency of mating increased from 95% on the fourth day to a full 100% on the fifth day of emergence. These findings emphasize that cotton seed bugs display sexual maturity and readiness to engage in mating as early as the third day following emergence.

As with many insect species [14], female *O. hyalinipennis* laid more eggs after mating (Figure 2). The current study strongly indicates that mating has an immediate impact on fecundity, leading to increased ovipositional activity. Mated females began laying eggs two days after mating and continued until day 34, depositing tenfold more eggs than virgin females (Figure 2). The stimuli prompting this behavior may arise from physical cues associated with copulation [15,16] and from a substance transferred within the spermatophore, observed in various insect orders such as *Drosophila melanogaster* [17], *Helicoverpa armigera* [18], and *Schistocerca gregaria* [19]. Furthermore, we noted instances of unmated females laying infertile eggs that failed to hatch. This phenomenon is not unique, as female insects of various species have been documented laying unfertilized eggs, including crickets *Velarifictorus aspersus* [15] and Western tarnished plant bug *Lygus hesperus* [16]. Notably, approximately 10% of the eggs laid by mated females remained unfertilized, at the same rate as the number of infertile eggs laid by unmated females (Figure 2). Further research is required to gain a deeper understanding of the reasons behind this behavior, particularly considering that egg production entails significant energy expenditure. The cumulative mean developmental time from egg-laying to adult under laboratory conditions at 26 °C and 75% RH was around 28 days (Figure 3). Of these, there are five nymphal stages, spanning ~23 days. The mean incubation period lasted for about 5.3 days. It is important to emphasize, however, that this developmental time can vary based on environmental factors such as temperature, relative humidity, and the availability of the host [10,20].

Mating positively influenced the survivorship and lifespan of adult cotton seed bugs. Remarkably, the lifespans of both male and female cotton seed bugs significantly increased as a result of mating (Figure 4). Previous studies have demonstrated that mating can either positively or negatively affect male and female survival in various insect species, including the fruit fly *Drosophila melanogaster* [21,22,23], lepidopteran species such as *Spodoptera littoralis* [24], and hematophagous insects such as *Aedes aegypti* [25] and the bed bug *Cimex lectularius* [26]. Further studies are required to gain comprehensive understanding of the effects of multiple mating events on the survivorship and lifespan of adult cotton seed bugs, considering that this study focused on a single mating event. As previously described, nymphs typically require seeds for their development, while adults consume a combination of seeds, leaves, and young stems [10,27]. In our laboratory conditions, however, we made an important observation: all nymphal stages exhibited a tendency to aggregate and feed on the green beans, the sole food source provided alongside cotton seeds and water. Notably, we observed a higher mortality rate among nymphs when green beans were absent from their diet. While it is well established that cotton seeds and other natural host plant seeds (mallows) are vital for *O. hyalinipennis* development and reproduction [10], the role of green beans as a supplementary food source for *O. hyalinipennis* growth had not previously been documented. The *O. hyalinipennis* poses a substantial risk to U.S. agricultural productivity due to the availability of suitable hosts and climate conditions, as well as the absence of natural predators that could restrict its population growth and spread, as well as damage. Since 1984, cotton seed bugs have been intercepted more than 600 times on various commodities at U.S. ports of entry. Understanding the life cycle and reproductive mechanisms of this pest is crucial for the formulation of an effective strategy for early detection, monitoring, control, containment, or eradication. The nymphal stages are the most vulnerable period. During this time, cotton seed bugs are more susceptible to different pest management practices, e.g., biological controls or the application pesticides, to break the life cycle of this pest. The critical insights gained in this study regarding factors that influence *O. hyalinipennis* survivor, reproduction, and abundance are expected to play a significant role in the development of control methods, ultimately improving crop yield while reducing synthetic chemical usage and safeguarding U.S. agriculture.

## 5. Conclusions

In conclusion, our study sheds light on the reproductive behavior and developmental dynamics of the cotton seed bug, *Oxycarenus hyalinipennis*. We found that the timing of reproductive behaviors in adult cotton seed bugs, including latency to copulate, copulation frequency, and copulation duration, is closely linked to their age since emergence, with mating occurring as early as the third day. This study demonstrates a significant immediate impact of mating on the egg-laying behavior of *O. hyalinipennis* females, resulting in heightened ovipositional activity, potentially influenced by various stimuli associated with copulation and sperm transfer. We also recorded the quantity of eggs laid by unmated and mated females during their initial oviposition cycle and post emergence, respectively. Mated females produced 10-fold more eggs than unmated females. Additionally, we observed that mating positively impacts the lifespans of both male and female cotton seed bugs, highlighting the potential benefits of mating in this species, which warrants further investigation. The cumulative developmental timeline from egg to adult was approximately 29 days under our experimental conditions. Furthermore, we also observed that unmated females lay infertile eggs, a behavior that necessitates further investigation. Overall, our findings contribute to a better understanding of the biology and ecology of this invasive pest and open avenues for future research in this field for the development of innovative IPM strategies.

## Figures and Tables

**Figure 1 insects-15-00065-f001:**
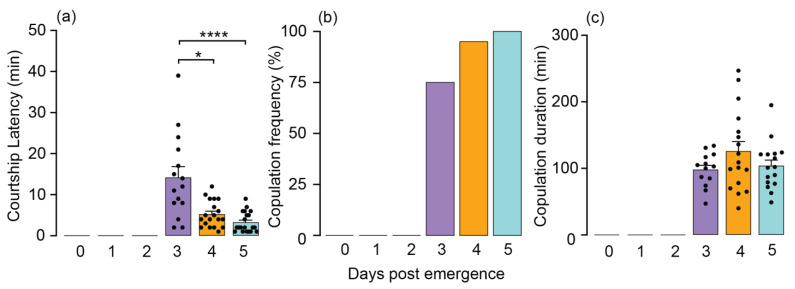
Courtship latency, copulation frequency, and copulation duration in *Oxycarenus hyalinipennis* males and females (*n* = 20). (**a**) Three-day-old adults displayed the longest latency to copulate (mean ± SEM: 14.20 ± 2.62), whereas the four- and five-day-old adults exhibited the shortest mean latency (mean ± SEM: 5.263 ± 0.73, and 3.30 ± 0.55, respectively). Copulation latencies are significant at **** *p*  <  0.0001 and * *p*  =  0.0262, according to Kruskal–Wallis and Dunn’s post hoc tests. Each dot within the bar indicates a replication. (**b**) Mating frequencies exhibited a significant surge starting from day 3 after emergence. Both male and female cotton seed bugs below the age of 2 days displayed reproductive inactivity. (**c**) Adults aged three, four, and five days old showed no effect on copulation duration (mean ± SEM: 98.31 ± 7.09, 126 ± 14.75, and 104.1 ± 8.94, respectively). Each dot within the bar indicates a replication.

**Figure 2 insects-15-00065-f002:**
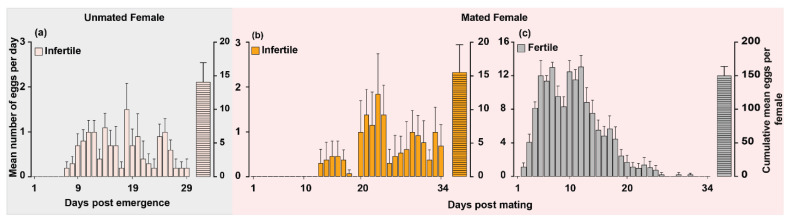
The egg-laying patterns of unmated and mated female *Oxycarenus hyalinipennis*. (**a**) Newly emerged unmated females were observed daily for egg-laying until it ceased, and the eggs were counted and monitored for development (mean ± SEM: 14.20 ± 2.913, *n*  =  10). (**b**,**c**) A five-day-old unmated male and female were paired for 24 h. Afterward, the male was removed, and the female was left in the container with a cotton seed for oviposition. The eggs were counted and monitored daily to record the number of infertile (**b**) and fertile (**c**) eggs. Bars show the mean daily eggs laid per female and the cumulative mean number of eggs oviposited per female (mean ± SEM: 15.54 ± 4.07, 151.8 ± 13.64, respectively; *n*  =  13).

**Figure 3 insects-15-00065-f003:**
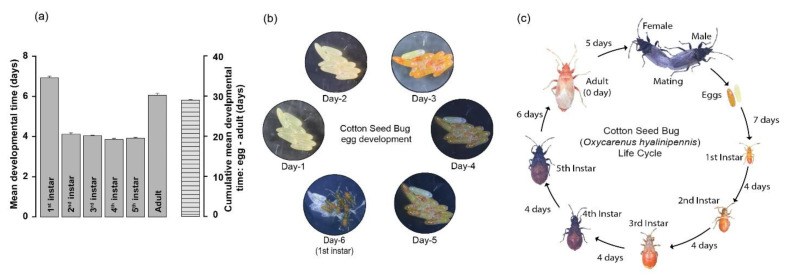
Developmental times of newly mated female *O. hyalinipennis* and life stages from egg to adult. Adult females post mating and eggs were monitored daily for the developmental time. (**a**) Bars show the mean developmental time (days) per female and the cumulative mean developmental time (days) from egg to adult per female (mean ± SEM: 28.96 ± 0.17, *n* = 50). (**b**) *O. hyalinipennis* egg development stages from egg to first instar nymphs. Notably, the eggs undergo color changes, transitioning from white to yellow, and ultimately to orange or pink. (**c**) *O. hyalinipennis* developmental stages encompass the progression from egg to adult: mature adults, eggs, five nymphal stages, and a newly emerged adult. Newly emerged adults are pale pink but rapidly turn brown and black.

**Figure 4 insects-15-00065-f004:**
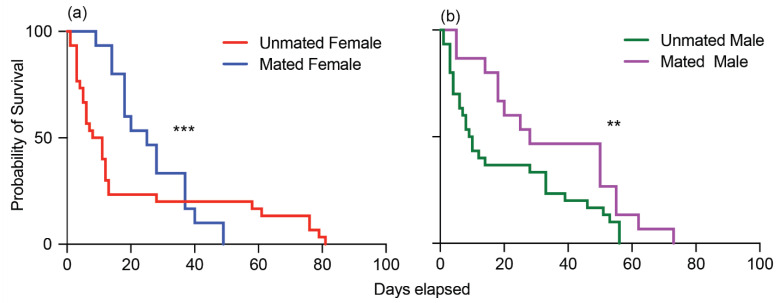
Effects of mating on the survival of male and female *Oxycarenus hyalinipennis*: (**a**) unmated female (red) and mated female (blue); (**b**) unmated male (green) and mated male (purple). Differences in the survival curves are significant (*** *p* < 0.0003, ** *p* < 0.003) according to the Gehan–Breslow–Wilcoxon test.

## Data Availability

The data presented in this study are available on request from the corresponding author.

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
