# Peer review of "Reproductive Behavior and Development of the Global Insect Pest, Cotton Seed Bug Oxycarenus hyalinipennis"

_insects, 2024, doi:10.3390/insects15010065_

Round 1
Reviewer 1 Report
Comments and Suggestions for Authors
The manuscript presents an interesting study about the reproductive behavior and development of the Cotton Seed Bug. The experiments are well designed and analyzed, the results are presented and discussed in a solid and appropriate way.
I have only minor comments and suggestions:
Line 30: I found the formulation a bit confusing when reading the abstract. I think the formulation in line 299/300 is much better to understand.
Line 251/254: I guess it is figure 4a and 4b and not figure 3a and b. The figure 4 should be presented in the results section.
Line 231/259: SEM value is missing.
Optional suggestions:
Line 280/282: It would be very interesting to get any data presented here. Any information about individual variation in mating behavior of insects are rare and very useful.
Author Response
Dear Editor,
On behalf of my coauthors and myself, I am very pleased to submit a revised version of our manuscript entitled, “Reproductive Behavior and Development of the Global Insect Pest, Cotton Seed Bug Oxycarenus hyalinipennis”, for consideration for publication in Insects. We are pleased to note that both reviews of the originally resubmitted manuscript were generally favorable. We address each of the referee’s concerns as follows:
Editorial Board Member comments:
Your manuscript has been seen by two reviewers who agree that your study of the reproductive behavior of the cotton seed bug Oxycarenus hyalinipennis is of scientific merit and well conducted. Both reviewers also provided lists of important corrections and suggestions that I would like you to consider in a revision of your manuscript. If you decide to submit a revision, please also provide a "tracked changes" version of your resubmitted manuscript for easy comparison with the original submission, in addition to the point-by-point response documents to the reviewer's comments.
Last but not least, I would like you to include the scientific name Oxycarenus hyalinipennis in the title and to refer to Oxycarenus hyalinipennis as O. hyalinipennis through hout the manuscript. For one, scientific nomenclature has been created to enable unambiguous reference to specific species. In addition, I think it is equally important that both experts and broader audiences are familiar with the scientific names of important pest species.
Response: We appreciate your and the 2 reviewers’ careful reviews. We believe that we addressed all of the reviewers’ concerns and we trust these changes will now facilitate a timely acceptance of this work for publication in Insects.
Reviewer 1:
General comments
The manuscript presents an interesting study about the reproductive behavior and development of the Cotton Seed Bug. The experiments are well designed and analyzed, the results are presented and discussed in a solid and appropriate way.
Response: Thank you for your kind words regarding our manuscript; your feedback is invaluable and encourages us in our research endeavors.
Line 30: I found the formulation a bit confusing when reading the abstract. I think the formulation in line 299/300 is much better to understand.
Response: Thank you for your comments. We've made the necessary changes accordingly.
Line 251/254: I guess it is figure 4a and 4b and not figure 3a and b. The figure 4 should be presented in the results section.
Response: Response: Thank you for your comments. We've made the necessary changes accordingly.
Optional suggestions:
Line 280/282: It would be very interesting to get any data presented here. Any information about individual variation in mating behavior of insects are rare and very useful.
Response: We agree that individual variations in insect mating behavior are crucial. However, in this study, we solely measured the latency, frequency, and duration among different age groups, as demonstrated in Figure 1.
Reviewer 2 Report
Comments and Suggestions for Authors
This paper investigated the reproductive behavior and developmental dynamics of the cotton seed bug, Oxycarenus hyalinipennis. The overall structure of the article is complete and the amount of data is sufficient, which could help to suppress O. hyalinipennis population cases of severe infestations. However, there are still some problems with the article, requiring revisions before publication.
1. L10: “Cotton seed bug” change to “Cotton seed bug (CSB)”
2. L13-15: “U.S. cotton industry…. in the U.S.”- Font, font size error, please adjust
3. Abstract: suggest to add your research significance in the last part of Abstract
4. Introduction: When introducing the damage of CSB, the control techniques, methods and chemicals used for CSB can be described
5. Section 2.2.1: What are the methods of observation and recording mating behavior? Use a camera or a human eye record?
6. Section 2.2.3: “eggs were collected by placing cotton seeds in the adult CSB colony container for 24 hours” - How many CSB female adults do these eggs come from?
7. Section 3.1: “To evaluate copulation metrics, including latency, frequency, and duration…observing them for an hour.” - The duration of copulation was only observed for one hour? Please re-describe to avoid misunderstanding of the reader, and make corresponding changes in the materials and methods
8. Figure 1: The error lines need to be marked in Figure 1a and Figure 1b
9. Figure 2: There is too much repetition in the description of the method and the content described in the material and method section in the Figure Legend. Suggest to simplify or delete
10. Section 3.4: “Figure 3a” change to “Figure 4a”, “Figure 3b” change to “Figure 4b”
11. Figure 4: Position moved to the lower part of Section 3.4
12. L275-287: Whether there is an age-dependent effect on mating behavior in other insects like this study, please try to discuss
13. L331-334: Suggest to give specific prevention and control suggestions according to the experimental results of this article, such as the best prevention period and prevention methods
14. References: Latin names of species should be italicized. Please check and unify the format of references according to the requirements of the journal
Author Response
Dear Editor,
On behalf of my coauthors and myself, I am very pleased to submit a revised version of our manuscript entitled, “Reproductive Behavior and Development of the Global Insect Pest, Cotton Seed Bug Oxycarenus hyalinipennis”, for consideration for publication in Insects. We are pleased to note that both reviews of the originally resubmitted manuscript were generally favorable. We address each of the referee’s concerns as follows:
Editorial Board Member comments:
Your manuscript has been seen by two reviewers who agree that your study of the reproductive behavior of the cotton seed bug Oxycarenus hyalinipennis is of scientific merit and well conducted. Both reviewers also provided lists of important corrections and suggestions that I would like you to consider in a revision of your manuscript. If you decide to submit a revision, please also provide a "tracked changes" version of your resubmitted manuscript for easy comparison with the original submission, in addition to the point-by-point response documents to the reviewer's comments.
Last but not least, I would like you to include the scientific name Oxycarenus hyalinipennis in the title and to refer to Oxycarenus hyalinipennis as O. hyalinipennis through hout the manuscript. For one, scientific nomenclature has been created to enable unambiguous reference to specific species. In addition, I think it is equally important that both experts and broader audiences are familiar with the scientific names of important pest species.
Response: We appreciate your and the 2 reviewers’ careful reviews. We believe that we addressed all of the reviewers’ concerns and we trust these changes will now facilitate a timely acceptance of this work for publication in Insects.
Reviewer 2:
General comments
This paper investigated the reproductive behavior and developmental dynamics of the cotton seed bug, Oxycarenus hyalinipennis. The overall structure of the article is complete and the amount of data is sufficient, which could help to suppress O. hyalinipennis population cases of severe infestations. However, there are still some problems with the article, requiring revisions before publication.
Response: Thank you for your kind words regarding our manuscript; your feedback is invaluable and encourages us in our research endeavors.
L10: “Cotton seed bug” change to “Cotton seed bug (CSB)”
Response: Thank you for your comments. We've made the necessary changes accordingly.
L13-15: “U.S. cotton industry…. in the U.S.”- Font, font size error, please adjust
Response: Thank you for your comments. We've made the necessary changes accordingly.
Abstract: suggest to add your research significance in the last part of Abstract
Response: Thank you for your comments. We've made the necessary changes accordingly.
When introducing the damage of CSB, the control techniques, methods and chemicals used for CSB can be described
Response: Thank you for your comments. We've made the necessary changes accordingly.
Section 2.2.1: What are the methods of observation and recording mating behavior? Use a camera or a human eye record?
Response: Thank you for your comments. We've made the necessary changes accordingly.
Section 2.2.3: “eggs were collected by placing cotton seeds in the adult CSB colony container for 24 hours” - How many CSB female adults do these eggs come from?
Response: Thank you for your comments. We've made the necessary changes accordingly.
Section 3.1: “To evaluate copulation metrics, including latency, frequency, and duration…observing them for an hour.” - The duration of copulation was only observed for one hour? Please re-describe to avoid misunderstanding of the reader, and make corresponding changes in the materials and methods
Response: Thank you for your comments. We've made the necessary changes accordingly.
Figure 1: The error lines need to be marked in Figure 1a and Figure 1b
Response: Error lines were incorporated into Figure 1a and Figure 1c, whereas Figure 1b doesn't have an error line because the data is presented as a percentage of the total number of mating pairs for each age group. For instance, 15 out of 20 (75%) tested 3-day-olds successfully mated.
.
Figure 2: There is too much repetition in the description of the method and the content described in the material and method section in the Figure Legend. Suggest to simplify or delete
Response: Thank you for your comments. We've made the necessary changes accordingly.
Section 3.4: “Figure 3a” change to “Figure 4a”, “Figure 3b” change to “Figure 4b”
Response: Thank you for your comments. We've made the necessary changes accordingly.
Figure 4: Position moved to the lower part of Section 3.4
Response: Thank you for your comments. We've made the necessary changes accordingly.
L275-287: Whether there is an age-dependent effect on mating behavior in other insects like this study, please try to discuss
Response: Thank you for your comments. We've made the necessary changes accordingly.
L331-334: Suggest to give specific prevention and control suggestions according to the experimental results of this article, such as the best prevention period and prevention methods
Response: Thank you for your suggestion. We have added a sentence accordingly.
References: Latin names of species should be italicized. Please check and unify the format of references according to the requirements of the journal
Response: Thank you for your feedback. We've made the necessary changes to the references.
Round 2
Reviewer 2 Report
Comments and Suggestions for Authors
Can be accepted.